# Safety and Osteointegration of Titanium Screws Coated with a Fibroblast Growth Factor-2–Calcium Phosphate Composite Layer in Non-Human Primates: A Pilot Study

**DOI:** 10.3390/jfb14050261

**Published:** 2023-05-08

**Authors:** Yukei Matsumoto, Hirotaka Mutsuzaki, Yuki Hara, Katsuya Nagashima, Eriko Okano, Yohei Yanagisawa, Hiroshi Noguchi, Tadashi Sankai, Masashi Yamazaki

**Affiliations:** 1Department of Orthopaedic Surgery, University of Tsukuba, 1-1-1 Tennodai, Tsukuba 305-8575, Japan; 2Center for Medical Science, Ibaraki Prefectural University of Health Sciences, Ami 300-0394, Japan; 3Department of Orthopedic Surgery, Ibaraki Prefectural University of Health Sciences Hospital, Ami 300-0331, Japan; 4Tsukuba Primate Research Center, National Institutes of Biomedical Innovation, Health and Nutrition, 1-1 Hachimandai, Tsukuba 305-0843, Japan

**Keywords:** fibroblast growth factor-2, FGF-2–calcium phosphate, pedicle screws

## Abstract

Spinal instrumentation surgery for older patients with osteoporosis is increasing. Implant loosening may occur due to inappropriate fixation in osteoporotic bone. Developing implants that achieve stable surgical results, even in osteoporotic bone, can reduce re-operation, lower medical costs, and maintain the physical status of older patients. Fibroblast growth factor-2 (FGF-2) promotes bone formation; thus, coating pedicle screws with an FGF-2–calcium phosphate (FGF-CP) composite layer is hypothesized to enhance osteointegration in spinal implants. We designed a long-term implantation pilot study that estimated the safety and bone-forming efficacy of pedicle screws coated with an FGF-CP composite layer in cynomolgus monkeys. Titanium alloy screws, either uncoated (controls) or aseptically coated with an FGF-CP composite layer, were implanted in the vertebral bodies of six female adult cynomolgus monkeys (three monkeys per group) for 85 days. Physiological, histological, and radiographic investigations were performed. There were no serious adverse events, and no radiolucent areas were observed around the screws in either group. The bone apposition rate in the intraosseous region was significantly higher in the FGF-CP group than in the controls. Moreover, as analyzed by Weibull plots, the bone formation rate of the FGF-CP group exhibited a significantly higher regression line slope than the control group. These results demonstrated that there was significantly less risk of impaired osteointegration in the FGF-CP group. Our pilot study suggests that FGF-CP-coated implants could promote osteointegration, be safe, and reduce the probability of screw loosening.

## 1. Introduction

Spinal instrumentation surgery is a useful surgical method for spinal diseases, such as degenerative scoliosis, vertebral fractures, and spinal stenosis. With the growing older adult population, the number of patients with osteoporosis is increasing, along with the use of spinal instrumentation surgery to treat these patients [1,2]. Complications—such as implant loosening, vertebral fracture, and implant failure—frequently occur in patients with osteoporosis [3,4]. Polymethylmethacrylate is one of the materials used to augment the fixation force, but there is a risk of cement extravasation [5]. Therefore, it is important to develop implants that can contribute to stable outcomes and safety, even in older patients with osteoporosis.

We developed a combination product including a spinal instrument with an osteoconductive coating and biologically active agent; the pedicle screw comprised fibroblast growth factor-2–calcium phosphate (FGF-CP) composite layers. Fibroblast growth factor-2 (FGF-2) and bone morphogenic protein (BMP) are growth factors that promote osteogenesis. Recombinant human BMP has undergone clinical trials and is already used clinically in spinal instrumentation surgery [6,7]; however, BMP-induced osteogenesis is dose-dependent, and high-dose BMPs increase the likelihood of complications, such as cancer and radiculopathy due to ectopic bone formation [8,9,10]. Calcium phosphate (CP) coatings and bone substitutes are biocompatible [11,12,13], have immunomodulatory effects [14,15], support osteogenic differentiation of mesenchymal stem cells [16,17,18], and show osteoconduction and bone-bonding abilities [19,20,21,22]. Thus, a pedicle screw coated with osteoconductive CP and a growth factor in its optimal dose could prevent the screw from loosening owing to osteoconduction on the screw associated with enhanced bone formation.

FGF-2 promotes the cellular differentiation of progenitor cells to osteoblasts and osteocytes [23]; additionally, it was reported that local application of 0.8 and 2.4 mg of recombinant human FGF-2 (rh FGF-2) in gelatin hydrogel accelerates bone union in humans [24]. It was also previously reported that ceramic hydroxyapatite coated with FGF-CP composite layers accelerated osteogenesis in a rat model of cranial bone defect, and low-dose rhFGF-2 also increased osteogenesis [25]. Further, titanium screws coated with FGF-CP composite layers have been shown to reduce the risk of impaired bone apposition to the screw in a rabbit model, in which the titanium screws were loaded with 2.0–4.7 µg of rh FGF-2 in a Bradford assay [26]. However, these previous animal studies used non-primate species. In non-primate species, rh FGF-2 is a foreign body that can induce safety and efficacy profiles that are different from those in primate species. In addition, the optimal dose of growth factor depends on the species, tissue, and drug carriers (such as gelatin and CP). Thus, it is preferable to carry out in vivo pathological evaluations of the safety and efficacy of pedicle screws coated with FGF-CP composite layers in non-human primates in the same tissues.

A clinical trial of external fixation pins coated with FGF-CP composite layers for fractures of the distal radius demonstrated safety during short-term clinical use (39–45 days), as well as a tendency towards a reduced pin tract infection rate [27]. The reduced pin tract infection rate was attributed to the enhancing effects of FGF-2 on soft tissue formation. In contrast, spinal instrumentation implants are permanently placed in the body, and thus bone and soft tissues around the screw are affected by FGF-2 for a long time. Another clinical trial of a pedicle screw coated with FGF-CP composite layers for cervical spine instrumentation surgery demonstrated safety in a single-arm trial [28]. However, most clinical trials have not demonstrated pathological safety and osteointegration that affect osteogenesis and screw loosening-linked efficacies based on comparisons.

The purpose of this study was to evaluate the pathological safety and efficacy of long-term implantation for osteointegration in cynomolgus monkeys, a close human relative. We hypothesized that titanium screws coated with FGF-CP composite layers would be safe, improve osteointegration, and reduce the risk of impaired osteointegration in monkey spinal implants.

## 2. Materials and Methods

### 2.1. Animals and Ethics Approval

The animal experiments were conducted at the Tsukuba Primate Research Center, National Institutes of Biomedical Innovation, Health, and Nutrition, Tsukuba, Ibaraki, Japan. All personnel involved in animal experimentation at the institute received educational training regarding monkey handling and biosafety. The prospective and blinded comparative study protocol was approved by the ethics committees of the University of Tsukuba (17-431), Ibaraki Prefectural University of Health Sciences (R4-8), and Tsukuba Primate Research Center (DS40-7).

### 2.2. Preparation of Implants

After crowns were removed, Ti-6Al-4V screws (4.5 mm in diameter and 30 mm in length, RENG Spinal System; Tanaka Medical Instruments Co., Ltd., Tokyo, Japan) were cut into lengths of 20 mm using a midget cutter (MC-0020 MCC; Osaka, Japan). The screws were washed with 90% ethanol for 30 min in an ultrasonic cleaner and sterilized using ethylene oxide gas.

### 2.3. FGF-CP Composite Layer Coating

The screw coating procedure was aseptically performed in a clean bench. Moreover, the coating method was a slightly modified version of a previously reported technique [26]. The screws were immersed in a supersaturated calcium phosphate solution at 37 °C for 3 h, followed by immersion in a supersaturated calcium phosphate solution containing FGF-2 (4.0 μg/mL) at 37 °C for 48 h. These supersaturated calcium phosphate solutions were prepared by mixing clinically available products as follows: Meylon injection 7% (NaHCO_3_; Otsuka Pharmaceutical, Tokyo, Japan), water for injection (FUSO Pharmaceutical Industries, Tokyo, Japan), Klinisalz (KH_2_PO_4_; KYOWA CritiCare Co., Ltd., Tokyo, Japan), dipotassium phosphate corrective injection (K_2_HPO_4_; Terumo Corporation, Tokyo, Japan), Ringer’s solution, calcium chloride injection, normal saline (Otsuka Pharmaceutical), and Fiblast (FGF-2; Kaken Pharmaceutical Co., Ltd., Tokyo, Japan) [29].

The coated screws were washed with sterile water, dried in a vacuum dryer (FDU-1200; EYELA, Tokyo, Japan) for 2 h at 12.5 Pa, and kept aseptically with desiccant in sealed plastic bags. Four screws were manufactured at once; three were for the animal study, and one was for quality assurance tests. To assess quality, the coating layer was dissolved using a citric acid solution, and deposited calcium and phosphorus were measured using an inductively coupled plasma atomic emission spectrometer (SPS7800; Seiko Instruments Inc., Chiba, Japan). The mitogenic activity of FGF-2 in the solution was examined using a fibroblastic NIH3T3 cell proliferation assay, as previously described [27].

### 2.4. Animal Experiments

Six female adult cynomolgus monkeys (*Macaca fascicularis,* approximately 11–28 years old) were equally divided into uncoated (control) and FGF-CP-coated groups. The animals had no previous history of spinal disease. They were individually caged under a 12-h light/dark cycle at 25 ± 3 °C and 50–70% humidity. Food was offered once a day, and access to water was ad libitum. All monkeys were healthy according to physical and blood examinations and were maintained following the appropriate rules for animal care and management.

All surgeries were performed under general anesthesia with isoflurane (Isoflu; Zoetis Inc., Parsippany, New Jersey, United States), followed by 10 mg/kg ketamine hydrochloride (Ketalar, Daiichi Sankyo, Tokyo, Japan) and 0.8–1.0 mg/kg xylazine hydrochloride (Celactal; Bayer, Leverkusen, Germany). Cephalosporin, a prophylactic antibiotic, was intramuscularly injected (500 mg) on the day of surgery and each of the 2 days following surgery (250 mg).

A left anterior retroperitoneal approach was used, and the lateral side of the lumber vertebral bodies was exposed. Three screws were implanted into three vertebral bodies of each animal. The surgical wound was closed fascia sutured, the subcutaneous tissue was sutured with absorbable sutures, and the monkeys were allowed to graze immediately after surgery without external immobilization. Health and behavioral observations, blood tests, and radiography were performed 1 week before the operation and on days 8 (1 week after the operation), 29, 57, and 85. Adverse events assumed to be complications of surgery included findings of infection, inflammation due to an allergic reaction to the FGF-CP composite layer, remarkable deterioration of health, and the occurrence of new diseases and injuries. 

Weight loss of ≥25% in 1 week qualified as Scientists Center for Animal Welfare (SCAW) category D and was a criterion for discontinuation (Increasing Ethical Concerns for Non-human Species made by SCAW). Therefore, the veterinarian performed health checks every day. Magnetic resonance imaging of the lumbar spine was performed 1 week before the operation and on day 85.

### 2.5. Infection and Inflammation Findings

The presence of infection and inflammation was evaluated by blood tests for C-reactive protein (CRP) and white blood cell count (WBC) and observation of the surgical site. Infection was indicated by CRP ≥ 10.0 mg/dL and WBC ≥ 15,000/μL at 1 week after surgery or CRP ≥ 2.0 mg/dL and WBC ≥ 15,000/μL 1 month after the operation. Redness and/or loosening of the wound and drainage at 1 week were considered to strongly suggest wound infection.

### 2.6. Radiography

Anteroposterior and lateral radiography of the screws was performed on days 8, 29, 57, and 85. Moreover, three blinded orthopedic physicians evaluated the radiographs. On anteroposterior radiographs, a wide (>1 mm) radiolucent zone surrounding the screws, regardless of the length of lucency, was defined as radiographic loosening.

Bone mineral density was measured at an unscrewed vertebral body by microcom-puted tomography (LCT-100A, Nihon Ray Tech Corporation, Tokyo, Japan).

### 2.7. Preparation of Tissue Specimens

The animals were euthanized on day 85 using 120 mg/kg pentobarbital (Somnopentyl; Kyoritsu Seiyaku Corporation, Tokyo, Japan). The vertebral bodies and screws were fixed in 10% neutral-buffered formalin and embedded in methyl methacrylate resin, and non-decalcified hard tissue specimens were prepared. Vertebral body sections (40 μm) were obtained parallel to the screw hole and across the center of the screw using a micro-cutting machine (BS-300CP MEIWAFOSIS Corporation, Tokyo, Japan). After polishing to exclude scratches, the sections were stained with hematoxylin–eosin (HE) and toluidine blue (TB). The spinal cord and soft tissue around the screw head were then fixed with 10% neutral-buffered formalin and embedded in paraffin. Further, the embedded samples were sliced into 5 μm sections and stained with HE and Masson’s trichrome. The sections were randomized and blinded.

### 2.8. Histology of the Soft Tissue around the Screw and Spinal Cord

The blinded soft tissue sections were examined via light microscopy (BX-51; Olympus Corporation, Tokyo, Japan) and independently analyzed by three physicians. High-powered (400× magnification) fields (HPFs) were randomly selected and blinded to examine malignant (n = 50) and inflammatory cells (n = 5) visually. The definition of a tumor lesion was 10/50 HPF or more. Inflammatory cells were scored based on ISO 10993-6 standard (score 0 is 0/HPF, score 1 is 0–5/HPF, score 2 is 5–10/HPF, score 3 is heavy infiltrate, and score 4 is packed) for cell types classified as granulocytes, lymphocytes, monocytes, plasmacytes, or giant cells at the spinal cord were classified as inflammatory cells except for giant cells at the bone around screws, because the slice of the bone tissues was 40 μm [30,31]. Infection was defined as 5/HPF of the granulocytes or more [32,33].

### 2.9. Histological Evaluation of the Bone around the Screw

The blinded, HE-stained, non-decalcified hard tissue specimens were examined by three physicians using a VAN0X-T microscope (Olympus Corporation). Images were captured with a 12.5× CCD video camera (DP80; Olympus Corporation) and analyzed using Image J software (National Institutes of Health, Bethesda, MD, USA). Three physicians independently examined the images. For the histomorphometric analysis of bone tissue reaction to the screw, rectangular areas—covering a valley between screw threads within the vertebral body (with height 0.55 m and length 2.0 mm)—were set as regions of interest (ROIs). Each individual ROI (not individual animal) was regarded as an independent sample and was statistically analyzed owing to the ethical limitations of animal use. The justification for this statistical analysis has been described elsewhere [31]. Setting each individual ROI as an independent sample could enable us to evaluate the bone tissue reactions covering a broad variety of bone quality since the screw is in contact with both vertebral bodies rich in cancellus bone and the tissue rich in cancellus bone in the vicinity of the vertebral canal and the pedicle. Setting each individual ROI as an independent sample is similar in condition to other study designs implanting multiple samples in one bone, such as the femur or mandible of an animal. In each ROI, bone apposition and formation rates were calculated and averaged over the three physicians. The bone apposition rate is defined as the implant surface length with bone contact divided by the implant surface length. Moreover, the bone formation rate is defined as the bone area on the implant surface divided by the tissue area on the implant surface [31,32,33] (Figure 1).

### 2.10. Weibull Plot Analysis

The risk of impaired osteointegration was evaluated using Weibull plot analysis. Weibull plot analysis is used to evaluate the probability of failure and/or reliability of industrial products [34]. This analysis was previously used to demonstrate the reduced risk of impaired bone apposition to screws using FGF-CP composite layer coatings [26]. In our study, poor bone apposition or formation rate was regarded as treatment failure in the Weibull plot analysis. 

The Weibull equation is as follows: lnln(1/(1−S))=mln(σ)−mln(ξ),
where ln is natural log, *S* is the probability of failure, *m* is the Weibull parameter, σ is the bone apposition or formation rate, and ξ the scale parameter.

Thus, the plot of ln(*σ*) against lnln(1/(1 − *S*) yields a straight line with a slope of *m*. In this study, *S* was the probability of resulting in a bone apposition or formation rate at or less than *σ*. To make the Weibull plot, measured *σ* values were arranged in ascending order, such as *σ*_1_, *σ*_2_, *σ_j_*, and *σ_N_*, where *j* is the order of an individual *σ* value and *N* is the total number of measured *σ* values. Then *S* was derived via the median rank method using *S_j_* = (*j* − 0.3)/(*N* + 0.4). Data with *σ* = 0 were excluded, as ln(*σ*) is invalid. Plots of ln(*σ_j_*) against lnln(1/(1 − *S_j_*) provided the Weibull plots. Using the regression lines of the Weibull plots, the probability of impaired bone apposition and formation rates was calculated at various threshold values of *σ*. The calculated probability was compared between the control and FGF-CP-coated groups.

### 2.11. Statistical Analyses

The Student’s two-tailed *t*-test was used to evaluate statistically significant differences. The F test was used to determine statistically significant differences in dispersion in the frequency histogram, and a *p*-value < 0.05 was considered statistically significant.

## 3. Results

### 3.1. FGF-CP Composite Layer

The calcium and phosphorus contents of the FGF-CP composite layer were as follows: Ca, 220.5 ± 28.0 μg/screw; P, 99.2 ± 13.2 μg/screw. The FGF-CP composite layer was found to promote fibroblast proliferation (Table 1), showing that the FGF-2 retained its biological activity.

### 3.2. Animal Study

All animals were female, and there were no significant differences regarding age, blood test results, radiography and MRI of the lumbar spine, bone mineral density, or body weight (Table 2). All animals that underwent screw implantation—either uncoated or coated with FGF-CP composite layers—survived for 85 days and demonstrated transient weight loss. None exhibited weight loss exceeding 15%, which was the stopping criterion. In both groups, no blood tests were abnormal, and plain radiography revealed no transparent bone images or deviation around the implant (Figure 2).

### 3.3. Histopathological Findings

No mitotic cells were found in the spinal cord or vertebral bone around the coated or uncoated screws (Table 3). Moreover, no inflammatory cells with scores of 3 or 4 or inflammatory reactions for implants were observed regarding the spinal cord or vertebral bone around the coated or uncoated screws (Table 3, Table 4 and Table 5, Figure 3 and Figure 4).

The distribution of data points of bone apposition rates (BS/IS) for the FGF-CP group was spread toward the greater value compared with that for the control group (Figure 5a). As a result, the average value of bone apposition rate was significantly higher in the FGF-CP group (41.5 ± 22.9%, n = 47) than in the control group (31.3 ± 18.6%, n = 34; *p* = 0.03). The distribution of data points of bone formation rates (BV/TV) for the FGF-CP group was more localized in the vicinity of the average value than that for the control group (Figure 5b). The average values of bone formation rate had no significant difference (*p* = 0.17; Figure 5) between the FGF-CP (36.0 ± 11.7%, n = 47) and control (31.6 ± 15.0%, n = 34) groups.

There was a significant difference (*p* = 0.03) in average bone apposition rate between the FGF-CP (41.5 ± 22.9%, n = 47) and control groups (31.3 ± 18.6%, n = 34); however, there was no significant difference (*p* = 0.17) in average bone formation rate between the FGF-CP (36.0 ± 11.7%, n = 47) and control groups (31.6 ± 15.0%, n = 34).

The risk of impaired bone apposition and formation rates was analyzed using Weibull plots (Figure 6). The data points of plots for bone apposition rate appeared to be fitted with bent lines that had lower and higher slopes in the lower and higher σ regions, respectively. However, it could not be ruled out that the line with a lower slope in the lower σ region was an artifact (Appendix A). Thus, in the present study, the data points for each bone apposition rate Weibull plot were regressed tentatively with straight lines that reflect the overall tendency of the plot.

The slope of the regression line of bone formation rate Weibull plot was significantly greater in the FGF-CP group than in the control group, demonstrating noticeably less risk of impaired osteointegration.

In the bone apposition Weibull plot, two data entries were removed in the FGF-CP and control groups, as the *σ* values were zero. The slope of the regression line for the FGF-CP group (1.11) was higher than that for the control group, but it was not statistically significant (0.89) (*p* = 0.06). The regression line for the FGF-CP group is located at a lower level compared with that for the control group. Using the regression lines, the risk of an impaired bone apposition rate was calculated at various threshold values of bone apposition rate, which were selected arbitrarily (Table 6). For example, when impaired bone apposition was defined as a state at ≤3.2%, ≤1.4%, and ≤0.2% of the bone apposition rate, the probabilities of impaired bone apposition in the control group was 10%, 5%, and 1%, respectively. The corresponding probabilities of impaired bone apposition in the FGF-CP group were 4.4%, 1.8%, and 0.2%, respectively, which were remarkably lower than those in the control group (Table 6).

In the bone formation Weibull plot, the slope of the regression line for the FGF-CP group was significantly higher (3.51) than that for the control group (2.05) (*p* = 1.07 × 10^−25^). When the impaired bone formation was defined as a state at ≤12%, ≤8.5%, and ≤3.8% of the bone formation rate, the probabilities of impaired bone formation in the control group were 10%, 5%, and 1%, respectively. The corresponding probabilities of impaired bone formation in the FGF-CP group were 1.5%, 0.4%, and 0.03%, respectively, which were remarkably lower than those in the control group (Table 7).

## 4. Discussion

It is important to examine the safety and effectiveness of a medical device in a clinically appropriate environment. In spinal instrumentation surgery, mechanical stress on the postoperative implants concentrates on the upper or lower ends of the fixed vertebral body [35]. The screw at the end vertebral body is prone to loosening due to the stress concentration, which can cause screw failure and pseudarthrosis. To test new spinal implants, the porcine spine is frequently used as a model for the human spine [36]; however, since the spine is mainly vertically loaded in humans, baboons are suggested to be more appropriate models for spinal degeneration [37]. Moreover, growth factors such as FGF-2 show species specificity in their biological activity and bell-shaped dose response, where both an overdose and a deficient dose exhibit diminished or even no effectiveness [38,39]. The FGF-2 used in the present study is a recombinant human protein. Thus, in the present study, the safety and effectiveness of pedicle screws coated and uncoated with an FGF-CP composite layer were examined in vertebral bodies of cynomolgus monkeys, a close relative of humans. This study verified histological and cytological safety. Moreover, in order to verify bone fixation in clinical trials, it was necessary to investigate osteointegration in an in vivo study using cynomolgus monkeys.

The present study demonstrated that 4.0 μg/mL of FGF-2 in a supersaturated calcium phosphate solution results in the formation of FGF-CP composite layers that potentially promote osteointegration in primates. Previous FGF-CP composite layers prepared under similar conditions contained poorly crystalline apatite or amorphous calcium phosphate and had a Ca/P molar ratio in the range of 1.55–1.85 [40,41,42,43]. Since the Ca/P molar ratio (1.71 ± 0.02, Table 1) in this study fell within a similar range to the previous study, poorly crystalline apatite or amorphous calcium phosphate could also be formed in the present study. Previous studies have examined the efficacy of FGF-CP composite layers in rabbits and rats, where the same concentration of FGF-2 (4.0 μg/mL) was shown to exert the most beneficial osteogenic effect [25,26,40,41,44]. In this pilot study, it was suggested that titanium alloy screws coated with FGF-CP composite layers prepared using the same concentration of FGF-2 also promoted bone apposition to the screws and reduced the risk of impaired bone apposition and formation when compared with uncoated screws in primates. The promoting effect of FGF-CP composite layers on osteointegration results from both calcium phosphate and FGF-2. Nonetheless, the dose of FGF-2 in the FGF-CP composite layer in this study is likely to contribute to osteointegration and remain within the effective dose range in primates. It should be noted that the probability of impaired bone apposition to uncoated titanium screws was similar to that of calcium phosphate-coated screws without FGF-2 in a previous study [26].

The Weibull plot analysis regarding impaired bone apposition or formation as treatment failure demonstrated the potential efficacy of osteointegration in the FGF-CP group. The Weibull plots were fitted with straight lines (Figure 6). Theoretically, the greater the slope of the line is, the more constant the treatment outcome and the lower the probability of failure. Under the same slope, the lower the straight line is located, the lower the probability of failure. Since the regression line of bone apposition rate Weibull plot for the FGF-CP group passes at a lower level than that of the control group, the FGF-CP has a tendency to lower the probability of impaired bone apposition compared with the control. The reason why the regression line for the FGF-CP group passes at a lower level is that the distribution of bone apposition rates for the FGF-CP group was spread toward the greater value compared with that for the control group (Figure 5a). It should be noted that the spread of distribution toward the greater value leads to an increase in scale parameter ξ in case the slope is the same [45]. In the present study, the distribution of bone apposition rates for the FGF-CP group was spread toward the greater value to a level that the average bone apposition rate was significantly higher in the FGF-CP group than in the control group. The fact that the slope of the regression line of bone formation rate Weibull plot was significantly greater in the FGF-CP group (3.51) than in the control group (2.05; *p* = 1.07 × 10^−25^) proves that the FGF-CP leads to a more constant bone formation rate and a lower probability of impaired bone formation than the control.

The Weibull plot analysis quantitatively demonstrated that the probability of im-paired bone apposition or formation rate was remarkably lower in the FGF-CP group than in the control group. This risk was expressed as the probability (S) of obtaining a bone apposition or formation rate equal to or less than an arbitrarily selected bone apposition or formation rate (σ). For example, in the control group, the probability of the bone apposition or formation rates being equal to or less than 3.2% or 12%, respectively, was 10%. In contrast, in the FGF-CP group, the probability of a bone apposition rate being equal to or less than 3.2% was only 4.4%, while the probability of a bone formation rate being equal to or less than 12% was 1.5%. Therefore, these findings suggest that pedicle screws coated with an FGF-CP composite layer are more reliable than uncoated screws, even in primates, for the prevention of impaired osteointegration that causes screw loosening.

This pilot study of FGF-CP-coated spinal screw implants in cynomolgus monkeys demonstrated tentative local safety based on a long-term in vivo study. Neither malignant cells nor infection-mediated inflammatory cell infiltration was observed around the coated screws. A previous study reported that gamma-ray-sterilized screws coated with FGF-CP were safe in cynomolgus monkeys [31]. In this study, no infiltration of inflammatory cells scored 3 or 4 in ISO 109993-6 was observed in the spinal cord and vertebral bone around the coated and uncoated screws. Further, no adverse events—excess osteogenesis in the spinal canal or motor paralysis—were observed; however, some implants penetrated the spinal canal. The FGF-2 on the implant surface may have been released into the spinal canal. Moreover, although bone and fibrous tissue could be induced into the spinal canal by the effect of FGF-2 on the implant surface, histopathology performed 85 days after implantation suggested that osteointegration could be induced without causing abnormal morphology of the spinal cord and nerve tissue.

There were limitations in this study. Due to ethical reasons, the minimum number of individuals required to demonstrate statistical power was not achieved, nor were mechanical tests performed. Instead, each individual ROI was regarded as an independent sample in the histomorphometric study. Due to the minimum number of individuals, screws coated with a calcium phosphate composite layer without FGF-2 were not examined in this study. Furthermore, the animals did not have osteoporosis; thus, we did not demonstrate osteoconduction in osteoporosis. The optimal dose of FGF-2 for maximizing osteoconduction is yet to be clarified.

## 5. Conclusions

This pilot study of long-term implantation tests on FGF-CP-coated spinal screws confirmed their safety in non-human primates. Tentatively, the screws induced significantly higher osteointegration and a remarkably lower probability of impaired osteointegration than uncoated screws. Our data suggest that FGF-CP-coated spinal screws could reduce the probability of screw loosening and contribute to stable outcomes in spinal instrumentation surgery. However, further studies are required to verify the results of our study.

## Figures and Tables

**Figure 1 jfb-14-00261-f001:**
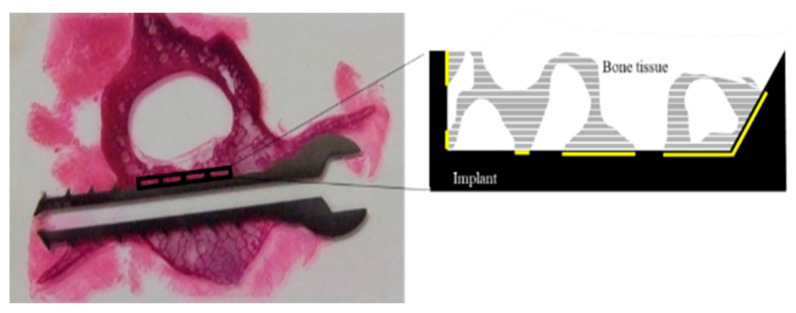
Histological evaluation of the bone around the screw. The region of interest (ROI) was defined as the rectangular area between a thread and the next thread in the vertebral body (square area: 0.55 mm × 2.0 mm). The bone apposition rate was defined as the ratio of the total length of the bone–screw interface line (yellow line), where the bone was in direct contact with the screw, to the length of the peripheral line of the screw. The bone formation rate was defined as the ratio of the total area of the mature bone tissue to the area of the ROI (grey horizontal line).

**Figure 2 jfb-14-00261-f002:**
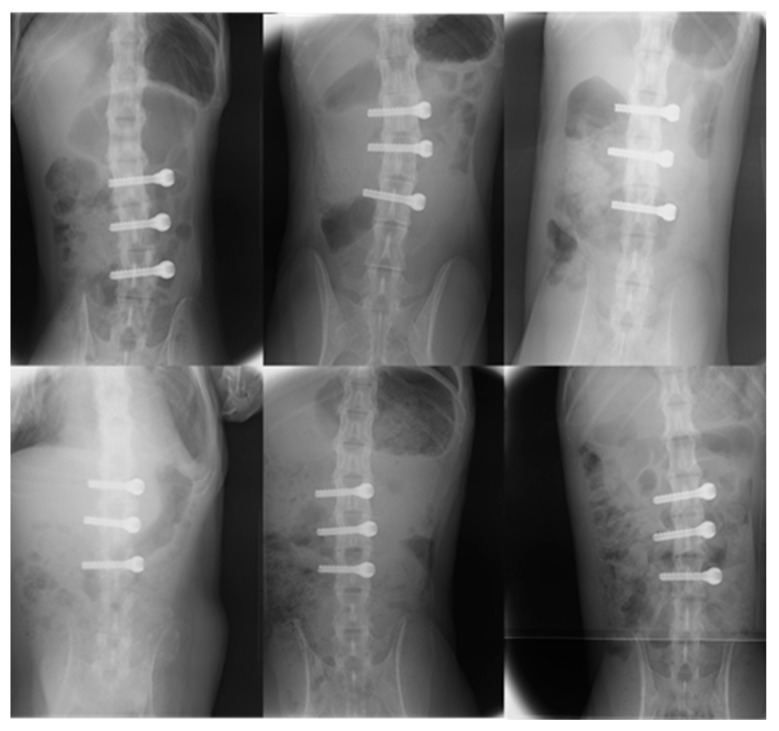
Radiographs (anteroposterior and lateral views) of the lumber spine 12 weeks after surgery (fibroblast growth factor-2–calcium phosphate (FGF-CP) group above, the control below). No lucent areas were observed around the screws in either group.

**Figure 3 jfb-14-00261-f003:**
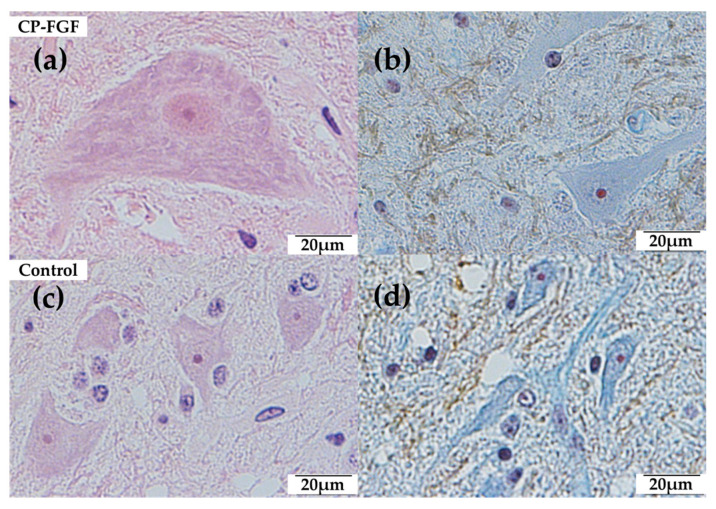
Histological sections of the spinal cord stained with hematoxylin and eosin (HE) and Masson’s trichrome (MT) 12 weeks after implantation. (**a**) HE (400×), (**b**) TB (400×), (**c**) HE (400×), (**d**) TB (400×). No mitotic cells and few inflammatory cells were identified in either group.

**Figure 4 jfb-14-00261-f004:**
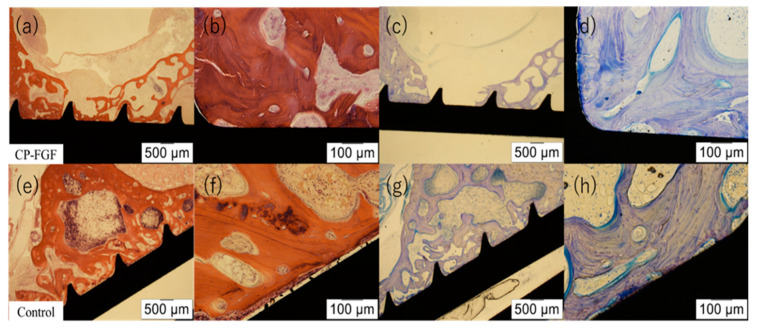
Histological sections of the lumbar vertebrae stained with hematoxylin and eosin (HE) and toluidine blue (TB) 12 weeks after implantation. (**a**) HE (12.5×), (**b**) HE (40×), (**c**) TB (12.5×), (**d**) TB (40×), (**e**) HE (12.5×), (**f**) HE (40×), (**g**) TB (12.5×), (**h**) TB (40×). Osteointegration was observed in both groups.

**Figure 5 jfb-14-00261-f005:**
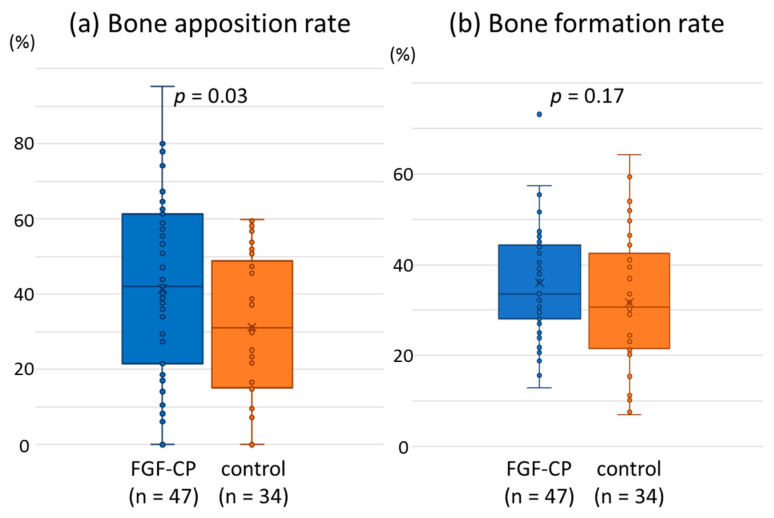
Bone apposition and bone formation rates.

**Figure 6 jfb-14-00261-f006:**
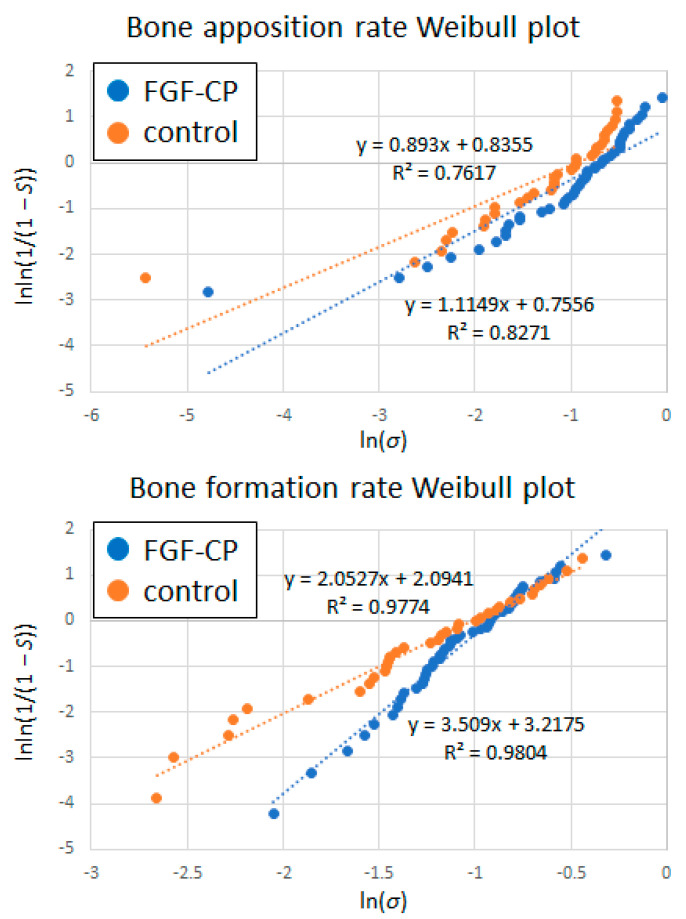
Weibull plots for bone apposition and formation rates.

**Table 1 jfb-14-00261-t001:** Characteristics of the fibroblast growth factor-2–calcium phosphate coating.

Components	Amount
Ca (µg/screw)	220.5 ± 28.0
P (µg/screw)	99.2 ± 13.2
Ca/P molar ratio	1.71 ± 0.02
FGF-2 activity	1.11 ± 0.02 (*p* = 0.005 relative to 1.0)

FGF-2, fibroblast growth factor-2.

**Table 2 jfb-14-00261-t002:** Baseline characteristics.

	FGF-CP	Control	*p*-Value
Number	3	3	
Age (years)	16.6	15.3	0.77
Liver dysfunction	1/3	2/3	
Impaired glucose tolerance	1/3	1/3	
X-ray	0/3	1/3 (scoliosis)	
MRI	0/3	1/3 (scoliosis)	
Bone mineral density (g/cm^2^)	0.410 ± 0.05	0.335 ± 0.08	0.33
Body weight (kg)	4.12 ± 0.68	3.54 ± 0.29	0.19

FGF-CP, fibroblast growth factor-2–calcium phosphate; MRI, magnetic resonance imaging.

**Table 3 jfb-14-00261-t003:** Mitotic cells of the spinal cord around the screws.

Vertebral Body		
Mitotic cell counts	<10/HPF	≥10/HPF
FGF-CP	1350	0
Control	1350	0
**Spinal cord**		
Mitotic cell counts	<10/HPF	≥10/HPF
FGF-CP	1650	0
Control	2100	0

FGF-CP, fibroblast growth factor-2–calcium phosphate.

**Table 4 jfb-14-00261-t004:** Inflammatory cells of the spinal cord around the screws.

Granulocyte					
Score	0	1	2	3	4
FGF-CP	164	1	0	0	0
Control	210	0	0	0	0
**Lymphocyte**					
Score	0	1	2	3	4
FGF-CP	111	53	1	0	0
Control	168	40	2	0	0
**Monocyte**					
Score	0	1	2	3	4
FGF-CP	165	0	0	0	0
Control	210	0	0	0	0
**Plasma cell**					
Score	0	1	2	3	4
FGF-CP	164	1	0	0	0
Control	210	0	0	0	0
**Giant cell**					
Score	0	1	2	3	4
FGF-CP	165	0	0	0	0
Control	210	0	0	0	0

FGF-CP, fibroblast growth factor-2–calcium phosphate.

**Table 5 jfb-14-00261-t005:** Inflammatory cells of the vertebral bone around the screws.

Inflammatory Cells, Except for Giant Cells					
Score	0	1	2	3	4
FGF-CP	101	34	0	0	0
Control	65	66	3	0	0
**Giant cells**					
Score	0	1	2	3	4
FGF-CP	134	1	0	0	0
Control	134	1	0	0	0

FGF-CP, fibroblast growth factor-2–calcium phosphate.

**Table 6 jfb-14-00261-t006:** Probability of impaired bone apposition at different bone apposition rates.

Bone Apposition Rate	Control	FGF-CP
≤3.2%	10%	4.4%
≤1.4%	5%	1.8%
≤0.2%	1%	0.2%

FGF-CP, fibroblast growth factor-2–calcium phosphate.

**Table 7 jfb-14-00261-t007:** Probability of impaired bone formation at different bone formation rates.

Bone Formation Rate	Control	FGF-CP
≤12%	10%	1.5%
≤8.5%	5%	0.4%
≤3.8%	1%	0.03%

FGF-CP, fibroblast growth factor-2–calcium phosphate.

## Data Availability

The datasets generated and/or analyzed during the current study are available from the corresponding author upon reasonable request.

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
