# Peer review of "Safety and Osteointegration of Titanium Screws Coated with a Fibroblast Growth Factor-2–Calcium Phosphate Composite Layer in Non-Human Primates: A Pilot Study"

_jfb, 2023, doi:10.3390/jfb14050261_

Round 1
Reviewer 1 Report
The manuscript "Safety and osteointegration of titanium screws coated with a fibroblast growth factor-2–calcium phosphate composite layer in 3 non-human primates" reports a histological evaluation of titanium screws functionalized with fibroblast growth factor-2 (FGF-2) and calcium phosphate precipitates. The findings presented in this manuscript are useful for understanding the effects of bioactive coating of titanium, and provide an important evidence of tissue responses to FGF-2-loaded titanium materials at the non-human primate level. The manuscript is well-written and demonstrates a fair evaluation. The limitation of this study is also clearly stated. Therefore, the reviewer believes that this manuscript can be acceptable for publication in Journal of Functional Biomaterials with some minor revisions.
Comment #1
The definition of bone formation rate is not very clear to the reviewer. The authors are suggested specifically providing the size (height × length) of ROI.
Comment #2
Although the expected functions of FGF-2 on bone regeneration are mentioned in Introduction, the major roles of calcium phosphate compounds are not clearly reviewed. The authors are suggested adding some literatures on this issue and introducing them.
Comment #3
The chemical nature of calcium phosphate coating is not clearly described except for elemental analyses on Ca and P. As well known, there are various types of calcium phosphate compounds such as hydroxyapatite, tricalcium phosphate, amorphous calcium phosphate etc., and the osteoconductivity is influenced by these composition. The authors are suggested describing the potential chemical composition of the calcium phosphate coating (by introducing previous papers, taking X-ray diffraction measurements, or speculating from existing data).
Comment #4
There are some erroneous and/or improper statements in Materials and Methods - 2.3. FGF–CP composite layer coating:
Klinisalz (H2PO4; KYOWA ...)
→ The chemical formula H2PO4 is mistaken. H3PO4, KH2PO4, or H2PO4- may be the intended one.
dipotassium phosphate corrective injection (PO43-; Terumo ...)
→ The correct chemical formula of dipotassium phosphate is K2HPO4.
The ionically denoted "PO43-" in the parenthesis is inconsistent with other ones in style.
calcium chloride injection
→ No chemical formula (CaCl2) is provided, which is inconsistent with other ones. The reviewer recommends describing the calcium source more clearly by denoting this chemical formula.
The authors are suggested correcting these notions.
Comment #5
The Weibull equation is presented as . However, the quantity ξ is not defined in the text. The authors are suggested defining all quantities if they need to be presented in this manuscript.
Comment #6
The Weibull distribution could show qualitatively different modes of failure rates S depending on the range of the Weibull parameter m: S (σ) drastically increases at lower σ region when m < 1, while S (σ) undergoes a drastic increase at higher σ region when m > 1. As shown in Fig. 6, the slopes (m) show both m > 1 and m < 1 cases, indicating different mechanisms of causing impaired bone formation. Therefore, the authors are suggested providing some comments on the mechanisms of impaired bone formation (apposition) with a remark on this qualitatively different distribution of failure rates.
Comment #7
Table 3 is not understandable to the reviewer. For example, the meaning of "0-9" and "10" are not described. The authors are suggested improving the captions.
Author Response
Responses to Reviewer #1 Thank you for your comments. As you kindly suggested, we improved the manuscript. Colored words below are parts of the text we revised according to your suggestions. Comment 1. The definition of bone formation rate is not very clear to the reviewer. The authors are suggested specifically providing the size (height × length) of ROI. Response 1. Thank you for your comment. According to your suggestion, we added the size (height × length) of ROI as follow. Previous Manuscript: 2.9 Histological evaluation of the bone around the screw “The blinded, HE-stained, non-decalcified hard tissue specimens were examined us-ing a VAN0X-T microscope (Olympus Corporation). Images were captured with a 12.5× CCD video camera (DP80; Olympus Corporation) and analyzed using Image J software (National Institutes of Health, Bethesda, MD, USA). Three physicians independently ex-amined the images. For the histomorphometric analysis of bone tissue reaction to the screw, rectangular areas—covering a valley between screw threads within the vertebral body—were set as regions of interest (ROIs). Each individual ROI (not individual animal) was regarded as an independent sample and was statistically analyzed due to the ethical limitations of animal use. Justification of this statistical treatment has been described elsewhere [18]. Setting each individual ROI as an independent sample could enable us to evaluate the bone tissue reactions covering a broad variety of bone quality since the screw is in contact with both vertebral bodies rich in cancellus bone and the tissue rich in can-cellus bone in the vicinity of the vertebral canal and the pedicle. Setting each individual ROI as an independent sample is similar in condition to other study designs implanting multiple samples in one bone, such as the femur or mandible of an animal. In each ROI, bone apposition and formation rates were calculated. The bone apposition rate is defined as the implant surface length with bone contact divided by the implant surface length. Moreover, the bone formation rate is defined as the bone area on the implant surface di-vided by the tissue area on the implant surface [18, 19, 20] (Figure 1).” Revised Manuscript: 2.9 Histological evaluation of the bone around the screw “The blinded, HE-stained, non-decalcified hard tissue specimens were examined by three physicians using a VAN0X-T microscope (Olympus Corporation). Images were captured with a 12.5× CCD video camera (DP80; Olympus Corporation) and analyzed using Image J software (National Institutes of Health, Bethesda, MD, USA). Three physicians independently examined the images. For the histomorphometric analysis of bone tissue reaction to the screw, rectangular areas—covering a valley between screw threads within the vertebral body (with height 0.55 m and length 2.0 mm)—were set as regions of interest (ROIs). Each individual ROI (not individual animal) was regarded as an independent sample and was statistically analyzed owing to the ethical limitations of animal use. The justification of this statistical analysis has been described elsewhere [31]. Setting each individual ROI as an independent sample could enable us to evaluate the bone tissue reactions covering a broad variety of bone quality since the screw is in contact with both vertebral bodies rich in cancellus bone and the tissue rich in cancellus bone in the vicinity of the vertebral canal and the pedicle. Setting each individual ROI as an independent sample is similar in condition to other study designs implanting multiple samples in one bone, such as the femur or mandible of an animal. In each ROI, bone apposition and formation rates were calculated and averaged over the three physicians. The bone apposition rate is defined as the implant surface length with bone contact divided by the implant surface length. Moreover, the bone formation rate is defined as the bone area on the implant surface divided by the tissue area on the implant surface [31-33] (Figure 1).” Previous Manuscript: 2. Materials and Methods, the annotation of Figure 1 “Histological evaluation of the bone around the screw.The region of interest (ROI) was de-fined as the rectangular area between a thread and the next thread in the vertebral body (square area). The bone apposition rate was defined as the ratio of the total length of the bone-screw in-terface line (yellow line), where the bone was in direct contact with the screw, to the length of the peripheral line of the screw. The bone formation rate was defined as the ratio of the area of the ROI to the total area of the mature bone tissue (grey horizontal line).” Revised Manuscript: 2. Materials and Methods, the annotation of Figure 1 “Histological evaluation of the bone around the screw. The region of interest (ROI) was de-fined as the rectangular area between a thread and the next thread in the vertebral body (square area: 0.55 mm x 2.0 mm). The bone apposition rate was defined as the ratio of the total length of the bone-screw interface line (yellow line), where the bone was in direct con-tact with the screw, to the length of the peripheral line of the screw. The bone formation rate was defined as the ratio of the area of the ROI to the total area of the mature bone tissue (grey hori-zontal line).” Comment 2. Although the expected functions of FGF-2 on bone regeneration are mentioned in Introduction, the major roles of calcium phosphate compounds are not clearly reviewed. The authors are suggested adding some literatures on this issue and introducing them. Response 2. Thank you for the comment. We have revised the introduction to add the major roles of calcium phosphate with addition of literatures (References 11-22). Previous Manuscript: 1. Introduction, 2nd paragraph “We developed a combination product including a spinal instrument with an osteocon-ductive coating and biologically active agent; the pedicle screw comprised fibroblast growth factor-2–calcium phosphate (FGF–CP) composite layers. Fibroblast growth factor-2 (FGF-2) and bone morphogenic protein (BMP) are growth factors that promote osteogene-sis. Recombinant human BMP has undergone clinical trials and is already used clinically in spinal instrumentation surgery [5,6]; however, BMP-induced osteogenesis is dose-dependent, and high-dose BMPs increase the likelihood of complications, such as cancer and radiculopathy due to ectopic bone formation [7-9].” Revised Manuscript: 1. Introduction, 2nd paragraph “We developed a combination product including a spinal instrument with an osteoconductive coating and biologically active agent; the pedicle screw comprised fibroblast growth factor-2–calcium phosphate (FGF–CP) composite layers. Fibroblast growth factor-2 (FGF-2) and bone morphogenic protein (BMP) are growth factors that promote osteogenesis. Recombinant human BMP has undergone clinical trials and is already used clinically in spinal instrumentation surgery [6,7]; however, BMP-induced osteogenesis is dose-dependent, and high-dose BMPs increase the likelihood of complications, such as cancer and radiculopathy due to ectopic bone formation [8-10]. Calcium phosphate (CP) coatings and bone substitutes are biocompatible [11-13], have immunomodulatory effects [14,15], support osteogenic differentiation of mesenchymal stem cells [16-18], and show osteoconduction and bone-bonding abilities [19-22]. Thus, a pedicle screw coated with osteoconductive CP and a growth factor in its optimal dose could prevent the screw from loosening owing to osteoconduction on the screw associated with enhanced bone formation.” Comment 3. The chemical nature of calcium phosphate coating is not clearly described except for elemental analyses on Ca and P. As well known, there are various types of calcium phosphate compounds such as hydroxyapatite, tricalcium phosphate, amorphous calcium phosphate etc., and the osteoconductivity is influenced by these composition. The authors are suggested describing the potential chemical composition of the calcium phosphate coating (by introducing previous papers, taking X-ray diffraction measurements, or speculating from existing data). Response 3. T Thank you for the comment which improves the quality of our manuscript. We have revised the manuscript to include the description about potential chemical composition and phase of the calcium phosphate coating by introducing previous papers and speculating from existing data. Previous Manuscript: 4. Discussion, 2nd paragraph “The present study demonstrated that 4.0 μg/ml of FGF-2 in a supersaturated calcium phosphate solution results in the formation of FGF–CP composite layers that potentially promote osteointegration in primates. Previous studies have examined the effectiveness of FGF–CP composite layers in rabbits and rats, where the same concentration of FGF-2 (4.0 μg/ml) was shown to exert the most beneficial osteogenic effect [12-13, 26-28]. In this study, titanium alloy screws coated with FGF–CP composite layers prepared using the same concentration of FGF-2 also promoted bone apposition to the screws and reduced the risk of impaired bone apposition and formation when compared with uncoated screws in primates. The promoting effect of FGF–CP composite layers on osteointegration results from both calcium phosphate and FGF-2. Nonetheless, the dose of FGF-2 in the FGF–CP composite layer in this study is likely to contribute to osteointegration and remain within the effective dose range in primates. It should be noted that the probability of impaired bone apposition to uncoated titanium screws was similar to that of calcium phos-phate-coated screws without FGF-2 in a previous study [13]. These results suggest safety and osteointegration based on pathological findings and clinical trials using spinal screws coated with FGF–CP composite layers. Moreover, we observed promising results when the layers were coated under the same conditions.” Revised Manuscript: 4. Discussion, 2nd paragraph “The present study demonstrated that 4.0 μg/ml of FGF-2 in a supersaturated calcium phosphate solution results in the formation of FGF–CP composite layers that potentially promote osteointegration in primates. Previous FGF–CP composite layers prepared under similar conditions contained poorly crystalline apatite or amorphous calcium phosphate and had a Ca/P molar ratio in the range of 1.55-1.85 [40-43]. Since the Ca/P molar ratio (1.71 ± 0.02, Table 1) in this study fell within similar range to the previous study, poorly crystalline apatite or amorphous calcium phosphate could also be formed in the present study. Previous studies have examined the effectiveness of FGF–CP composite layers in rabbits and rats, where the same concentration of FGF-2 (4.0 μg/ml) was shown to exert the most beneficial osteogenic effect [13,14, 27,28,31]. In this pilot study, it was suggested that titanium alloy screws coated with FGF–CP composite layers prepared using the same concentration of FGF-2 also promoted bone apposition to the screws and reduced the risk of impaired bone apposition and formation when compared with uncoated screws in primates. The promoting effect of FGF–CP composite layers on osteointegration results from both calcium phosphate and FGF-2. Nonetheless, the dose of FGF-2 in the FGF–CP composite layer in this study is likely to contribute to osteointegration and remain within the effective dose range in primates. It should be noted that the probability of impaired bone apposition to uncoated titanium screws was similar to that of calcium phosphate-coated screws without FGF-2 in a previous study [26]. These results suggest safety and osteointegration based on pathological findings and clinical trials using spinal screws coated with FGF–CP composite layers. Moreover, we observed promising results when the layers were coated under the same conditions.” Comment 4. There are some erroneous and/or improper statements in Materials and Methods - 2.3. FGF–CP composite layer coating: Klinisalz (H2PO4; KYOWA ...) → The chemical formula H2PO4 is mistaken. H3PO4, KH2PO4, or H2PO4- may be the intended one. dipotassium phosphate corrective injection (PO43-; Terumo ...) → The correct chemical formula of dipotassium phosphate is K2HPO4. The ionically denoted "PO43-" in the parenthesis is inconsistent with other ones in style. calcium chloride injection → No chemical formula (CaCl2) is provided, which is inconsistent with other ones. The reviewer recommends describing the calcium source more clearly by denoting this chemical formula. The authors are suggested correcting these notions. Response 4. Thank you for your comment. According to your suggestion, we corrected them as follow. Previous Manuscript: 2.3 FGF–CP composite layer coating, 1st paragraph “The screw coating procedure was aseptically performed on a clean bench. Moreover, the coating method was a slightly modified version of a previously reported technique [13]. The screws were immersed in a supersaturated calcium phosphate solution at 37°C for 3 h, followed by immersion in a supersaturated calcium phosphate solution containing FGF-2 (4.0 μg/mL) at 37°C for 48 h. These supersaturated calcium phosphate solutions were prepared by mixing clinically available products as follows: Meylon injection 7% (NaHCO3; Otsuka Pharmaceutical, Tokyo, Japan), water for injection (FUSO Pharmaceutical Industries, Tokyo, Japan), Klinisalz (H2PO4; KYOWA CritiCare Co., Ltd., Tokyo, Japan), dipotassium phosphate corrective injection (PO43; Terumo Corporation, Tokyo, Japan), Ringer’s solution, calcium chloride injection, normal saline (Otsuka Pharmaceutical), and Fiblast (FGF-2; Kaken Pharmaceutical Co., Ltd., Tokyo, Japan) [16].” Revised Manuscript: 2.3 FGF–CP composite layer coating, 1st paragraph “The screw coating procedure was aseptically performed on a clean bench. Moreover, the coating method was a slightly modified version of a previously reported technique [26]. The screws were immersed in a supersaturated calcium phosphate solution at 37°C for 3 h, followed by immersion in a supersaturated calcium phosphate solution containing FGF-2 (4.0 μg/mL) at 37°C for 48 h. These supersaturated calcium phosphate solutions were prepared by mixing clinically available products as follows: Meylon injection 7% (NaHCO3; Otsuka Pharmaceutical, Tokyo, Japan), water for injection (FUSO Pharmaceutical Industries, Tokyo, Japan), Klinisalz (KH2PO4; KYOWA CritiCare Co., Ltd., Tokyo, Japan), dipotassium phosphate corrective injection (K2HPO4; Terumo Corporation, Tokyo, Japan), Ringer’s solution, calcium chloride injection, normal saline (Otsuka Pharmaceutical), and Fiblast (FGF-2; Kaken Pharmaceutical Co., Ltd., Tokyo, Japan) [29].” Comment 5. The Weibull equation is presented as . However, the quantity ξ is not defined in the text. The authors are suggested defining all quantities if they need to be presented in this manuscript. Response 5. Thank you for the comment. We have defined all quantities and abbreviations used in the Weibull equation in the section 2.10 Weibull plot analysis. Previous Manuscript: 2.10 Weibull plot analysis, 2nd and 3rd paragraph “The Weibull equation is: lnln(1/(1 − S)) = mln(σ) − mln(ξ), where S is the probability of failure, m is the Weibull parameter, and σ the bone apposition or formation rate. Thus, the plot of ln(σ) against lnln(1/(1 − S ) yields a straight line with a slope of m. In this study, S was the probability of resulting in a bone apposition or formation rate at or less than σ. To design the Weibull plot, measured σ values were arranged in ascending order, such as σ1, σ2, , , σj, , , and σN, where j is the order of an individual σ value, and N is the total number of measured σ values. Then S was derived via the median rank method using Sj = (j–0.3)/(N+0.4). Data with σ = 0 were excluded, as ln(σ) is invalid. Plots of ln(σj) against lnln(1/(1 − Sj ) provided the Weibull plots. Using the regression lines of the Weibull plots, the probability of impaired bone apposition and formation rates was calculated at various threshold values of σ. The calculated probability was compared between the con-trol and FGF–CP-coated groups.” Revised Manuscript: 2.10 Weibull plot analysis, 2nd and 3rd paragraph “The Weibull equation is: lnln(1/(1 − S)) = mln(σ) − mln(ξ), where ln is natural log, S is the probability of failure, m is the Weibull parameter, σ is the bone apposition or formation rate, and ξ the scale parameter. Thus, the plot of ln(σ) against lnln(1/(1 − S ) yields a straight line with a slope of m. In this study, S was the probability of resulting in a bone apposition or formation rate at or less than σ. To make the Weibull plot, measured σ values were arranged in ascending or-der, such as σ1, σ2, , , σj, , , and σN, where j is the order of an individual σ value, and N is the total number of measured σ values. Then S was derived via the median rank method us-ing Sj = (j–0.3)/(N+0.4). Data with σ = 0 were excluded, as ln(σ) is invalid. Plots of ln(σj) against lnln(1/(1 − Sj ) provided the Weibull plots. Using the regression lines of the Weibull plots, the probability of impaired bone apposition and formation rates was calculated at various threshold values of σ. The calculated probability was compared between the control and FGF–CP-coated groups.” Comment 6. The Weibull distribution could show qualitatively different modes of failure rates S depending on the range of the Weibull parameter m: S (σ) drastically increases at lower σ region when m < 1, while S (σ) undergoes a drastic increase at higher σ region when m > 1. As shown in Fig. 6, the slopes (m) show both m > 1 and m < 1 cases, indicating different mechanisms of causing impaired bone formation. Therefore, the authors are suggested providing some comments on the mechanisms of impaired bone formation (apposition) with a remark on this qualitatively different distribution of failure rates. Response 6. Thank you for the very nice comment. The reviewer made us recognize the important difference in the slopes (m) in Figure 6. Prior to responding to this very nice comment, we have corrected an error in previous Figure 6 in making bone apposition rate Weibull plot for control. In previous calculation of the probability, Sj = (j–0.3)/(N+0.4), a wrong vale of 47 for N was used instead of the correct value of 34. The correct value of 34 for N gives a bone apposition rate Weibull plot for control as shown in Figure R1 right. The correct slope (m) is not 0.7732 but 0.893. Figure R1. Bone apposition rate Weibull plot for the control group. Previous wrong one (left), and corrected one (right). Then we would like to go back to the issue of the slopes (m) showing both m > 1 and m < 1 as follows. The m>1 and m1 and m1 (Figure R3). Figure R3. Bone apposition rate Weibull plot for the control group based on the assumption that the judgement of σ = 0 made by the two physician is false and that made by the third physician is true. On the other hand, the Weibull plot of Figure R1 right can be regressed by two straight lines with slopes (m) of m>1 and m1 (Figure R5 lower right). Figure R5. Bone apposition rate Weibull plot for FGF-CP. Upper: a plot fitted with two straight lines in the lnσ-1.67) ranges. Lower left: a plot based on the assumption that the judgement of σ = 0 made by the two physician is true and that made by the third physician is false on the data point corresponding to the lowest σ value. Lower right: a plot based on the assumption that the judgement of σ = 0 made by the two physician is false and that made by the third physician is ture on the data point corresponding to the lowest σ value. Therefore, there are two possibilities: The first one is that the slope (m) of m1. Thus, the bone formation rate Weibull plot indicates only one mechanism of causing impaired bone formation with a slope of m>1. On the basis of reanalysis of the data set of σ and their Weibull plots, as mentioned above, we have provided comments on whether the slopes are m > 1 or m < 1 with remarks on both qualitatively different failure modes and artifact in Supplementary materials, Results, and Discussion. In addition, the significant difference in previous slopes in bone apposition rate Weibull plots (0.77 for control vs. 1.11 for FGF-CP; p = 1.78×10−3 in two-tailed t-test) has disappeared after the correction of data for control (Figure R1; 0.89 for control vs. 1.11 for FGF-CP; p = 0.06 in two-tailed t-test). However, the FGF-CP group still has a tendency of reducing the risk of impaired bone apposition rate compared with the control group. In order to discuss the reduction of risk shown in Weibull plots, we have changed the bar graphs in Figure 5 into box-and-whisker plots, and revised Results and Discussion parts. Previous Manuscript: 3.3 Histopathological findings, 2nd – 3rd paragraph “The bone apposition rate (BS/IS) was significantly higher in the FGF–CP group (41.5 ± 22.9%, n = 47) than in the control group (31.3 ± 18.6%, n = 34; p = 0.03; Figure 6). The bone formation rates (BV/TV) were 36.0 ± 11.7% (n = 47) in the FGF–CP group and 31.6 ± 15.0% (n = 34) in the control group, which was not significantly different (p = 0.17; Figure 5). Figure 5. Bone apposition and bone formation rates. There was a significant difference in the bone apposition rate between the fibroblast growth factor-2–calcium phosphate (FGF–CP) and control groups; however, there was no significant difference in bone formation rate. The risk of impaired bone apposition and formation rates was analyzed using the Weibull plots (Figure 6). Regarding the bone apposition rate, the slope of the regression line for the bone apposition rate was significantly higher in the FGF–CP group (1.11) than in the control group (0.77) (p = 1.78×10−3). Two data entries were removed in the FGF–CP and control groups, as the σ values were zero. Using the regression lines, the risk of an impaired bone apposition rate was calculated at various threshold values of bone apposition rate, which were selected arbitrarily (Table 6). For example, when impaired bone apposition was defined as a state at ≤1.2%, ≤0.5%, and ≤0.1% of the bone apposition rate, the probability of impaired bone apposition in the control group was 10%, 5%, and 1%, respectively. The corresponding probabilities of impaired bone apposition in the FGF–CP group were 0.5%, 0.2%, and 0.02%, respectively, which was remarkably lower than those in the control group (Table 6). Regarding the bone formation rate, the slope of the regression line for the bone formation rate was significantly higher in the FGF–CP group (3.51) than in the control group (2.05) (p = 2.00 × 10−25). When the impaired bone formation was defined as a state at ≤12%, ≤9.1%, and ≤4.2% of the bone formation rate, the probabilities of impaired bone formation in the control group were 10%, 5%, and 1%, respectively. The corresponding probabilities of impaired bone formation in the FGF–CP group were 1.9%, 0.5%, and 0.03%, respectively, which were remarkably lower than those in the control group (Table 7). Figure 6. Weibull plots for bone apposition and formation rates The slope of the regression line was noticeably greater for the fibroblast growth factor-2–calcium phosphate (FGF–CP) group than that for the control group, demonstrating significantly less risk of impaired osteointegration.” Revised Manuscript: 3.3 Histopathological findings, 2nd – 4th paragraph “The distribution of data points of bone apposition rates (BS/IS) for the FGF–CP group was spread toward the greater value compared with that for the control group (Figure 5a). As a result, the average value of bone apposition rate was significantly higher in the FGF–CP group (41.5 ± 22.9%, n = 47) than in the control group (31.3 ± 18.6%, n = 34; p = 0.03). The distribution of data points of bone formation rates (BV/TV) for the FGF–CP group was more localized in the vicinity of the average value than that for the control group (Figure 5b). The average values of bone formation rate had no significant difference (p = 0.17; Figure 5) between the FGF–CP (36.0 ± 11.7%, n = 47) and control (31.6 ± 15.0%, n = 34) groups. Figure 5. Bone apposition and bone formation rates. There was a significant difference (p = 0.03) in average bone apposition rate between the FGF–CP (41.5, n=47) and control groups (31.3, n=34); however, there was no significant difference (p = 0.16) in average bone formation rate between the FGF–CP (36.0, n=47) and control groups (31.6, n=34). The risk of impaired bone apposition and formation rates was analyzed using the Weibull plots (Figure 6). The data points of plots for bone apposition rate appeared to be fitted with bent lines that had lower and higher slopes in the lower and higher σ regions, respectively. However, it was unable to rule out that the line with a lower slope in the lower σ region was an artifact (supplementary materials). Thus, in the present study, the data points for each bone apposition rate Weibull plot were regressed tentatively with straight lines that reflect the overall tendency of the plot. In the bone apposition Weibull plot, two data entries were removed in the FGF–CP and control groups, as the σ values were zero. The slope of the regression line for the FGF–CP group (1.11) was higher than that for the control group, but it was not statistically significant (0.89) (p = 0.06). The regression line for the FGF–CP group is located at a lower level compared with that for the control group. Using the regression lines, the risk of an impaired bone apposition rate was calculated at various threshold values of bone apposition rate, which were selected arbitrarily (Table 6). For example, when impaired bone apposition was defined as a state at ≤3.2%, ≤1.4%, and ≤0.2% of the bone apposition rate, the probability of impaired bone apposition in the control group was 10%, 5%, and 1%, respectively. The corresponding probabilities of impaired bone apposition in the FGF–CP group were 4.4%, 1.8%, and 0.2%, respectively, which was remarkably lower than those in the control group (Table 6). In the bone formation Weibull plot, the slope of the regression line for the FGF–CP group was significantly higher (3.51) than that for the control group (2.05) (p = 1.07 × 10−25). When the impaired bone formation was defined as a state at ≤12%, ≤8.5%, and ≤3.8% of the bone formation rate, the probabilities of impaired bone formation in the control group were 10%, 5%, and 1%, respectively. The corresponding probabilities of impaired bone formation in the FGF–CP group were 1.5%, 0.4%, and 0.03%, respectively, which were remarkably lower than those in the control group (Table 7). Figure 6. Weibull plots for bone apposition and formation rates The slope of the regression line of bone formation rate Weibull plot was greater in the FGF–CP group than in the control group, demonstrating noticeably less risk of impaired osteointegration.” Previous Manuscript: 4. Discussion, 3rd paragraph “Regarding the potential effectiveness of osteointegration, this study demonstrated a remarkably reduced risk of impaired osteointegration in the FGF–CP group. Regarding impaired bone apposition or formation as treatment failure, the Weibull plot analysis quantitatively demonstrated that the probability of impaired bone apposition or formation rate was remarkably lower in the FGF–CP group than in the control group. The Weibull plot provided straight lines. Theoretically, the greater the slope of the line, the more constant the treatment outcome and the lower the probability of failure. The fact that the slope of the regression line was greater in the FGF–CP group (1.11) than in the control group (0.77; p = 1.78×10−3) demonstrated that there was significantly less risk of impaired bone apposition or formation in the FGF–CP group than in the control group. This risk was ex-pressed as the probability (S) of obtaining a bone apposition or formation rate equal to or less than an arbitrarily selected bone apposition or formation rate (σ). For example, in the control group, the probability of a bone apposition or formation rate being equal to or less than 1.2% or 12%, respectively, was 10%. In contrast, in the FGF–CP group, the probability of a bone apposition rate being equal to or less than 1.2% was only 0.5%, while the probability of a bone formation rate being equal to or less than 12% was 1.9%. The reduced risk of impaired bone apposition in the FGF–CP group was supported by the narrower frequency distribution for bone apposition rate compared to the control group (the F test, p = 0.06: data not shown). Even on average, the bone apposition rate in the FGF–CP group was significantly higher in the control group. Therefore, these findings suggest that pedicle screws coated with an FGF–CP composite layer are more reliable than uncoated screws, even in primates, for the prevention of impaired osteointegration that causes screw loosening.” Revised Manuscript: 4. Discussion, 3rd -4th paragraph “The Weibull plot analysis regarding impaired bone apposition or formation as treatment failure demonstrated the potential efficacy of osteointegration in the FGF–CP group. The Weibull plots were fitted with straight lines (Figure 6). Theoretically, the greater the slope of the line is, the more constant the treatment outcome and the lower the probability of failure. Under the same slope, the lower the straight line is located, the lower the probability of failure. Since the regression line of bone apposition rate Weibull plot for the FGF–CP group passes at a lower level than that of the control group, the FGF–CP has a tenden-cy to lower the probability of impaired bone apposition compared with the control. The reason why the regression line for the FGF–CP group passes at a lower level is that the distribution of bone apposition rates for the FGF–CP group was spread toward the greater value compared with that for the control group (Figure 5a). It should be noted that the spread of distribution toward the greater value leads to an increase in scale parameter ξ in case the slope is the same [45]. In the present study, the distribution of bone apposition rates for the FGF–CP group was spread toward the greater value to a level that the average bone apposition rate was significantly higher in the FGF–CP group than in the control group. The fact that the slope of the regression line of bone formation rate Weibull plot was significantly greater in the FGF–CP group (3.51) than in the control group (2.05; p = 1.07 × 10−25) proves that the FGF–CP leads to the more constant bone formation rate and the low-er probability of impaired bone formation than the control. The Weibull plot analysis quantitatively demonstrated that the probability of impaired bone apposition or formation rate was remarkably lower in the FGF–CP group than in the control group. This risk was expressed as the probability (S) of obtaining a bone apposition or formation rate equal to or less than an arbitrarily selected bone apposition or formation rate (σ). For example, in the control group, the probability of the bone apposition or formation rates being equal to or less than 3.2% or 12%, respectively, was 10%. In contrast, in the FGF–CP group, the probability of a bone apposition rate being equal to or less than 3.2% was only 4.4%, while the probability of a bone formation rate being equal to or less than 12% was 1.5%. Therefore, these findings suggest that pedicle screws coated with an FGF–CP composite layer are more reliable than uncoated screws, even in primates, for the prevention of impaired osteointegration that causes screw loosening.” Comment 7. Table 3 is not understandable to the reviewer. For example, the meaning of "0-9" and "10" are not described. The authors are suggested improving the captions. Response 7. Thank you for your comment. According to your suggestion, we corrected the Table 3 as follow. Previous Manuscript: Table 3 Table 3. Mitotic cells of the spinal cord around the screws. Vertebral body Mitotic cells 0-9 ≥10 FGF-CP 1350 0 Control 1350 0 Spinal cord Mitotic cells 0-9 ≥10 FGF-CP 1650 0 Control 2100 0 FGF-CP, fibroblast growth factor-2–calcium phosphate Revised Manuscript: Table 3 Table 3. Mitotic cells of the spinal cord around the screws. Vertebral body Mitotic cell counts
Reviewer 2 Report
The objective of this manuscript was to explore and analyze the safety and osteointegration of titanium screws coated with a fibroblast growth factor-2–calcium phosphate composite layer in non-human primates.
Title: considering the small number of animals, also due to the ethical issues related to the 3R, I believe that the title of the work should be changed. It might be worth adding that this is a pilot study.
Abstract: Even in the abstract it should be emphasized that this is a pilot study. Also, I think it might be useful to introduce the issue of spinal fusion surgery already in the abstract.
Introduction:
-all the first part of the introduction (lines 32-38) should be implemented, focusing on the clinical problems currently present for this type of surgery.
-lines 55-57 “clinical applications require preclinical in vivo pathological evaluations of safety and efficacy in non-human primates”. What ISO standard is it? safety, efficacy and osseointegration are always evaluated in small animals.
Materials and Methods
Animal experiments:
Lines 122: ‘Three screws were implanted into three vertebral bodies of each animal. The surgical wound was closed in a standard manner’. Please explain.
Preparation of tissue specimens:
Line 148: ‘non-decalcified hard tissue specimens were prepared.’ How? Please explain the hard resin used and how you obtain 40 µm sections.
Line 151: microcomputed tomography. Please explain the acquisition and reconstruction methods. I think a dedicated paragraph would be needed.
Histological evaluation of the bone around the screw
Have you calculated the BIC?
Figure 1: is it a scan or histology? I believe it is a macro scan.
Figure 3: Histological sections are not so clear.
What about micro-CT results?
Discussion
Line 303: ‘medical product’. Please replace with medical device.
Line 310: Please add references.
I believe that given the low number of animals used, the discussion should be modified, trying to underline that the results are preliminary and that their value is in relation to the number of animals. The use of 3 animals per experimental group does not allow us to be so conclusive. Thus, the discussion should be substantially modified.
Author Response
Responses to Reviewer #2 Thank you for your comments. As you kindly suggested, we improved the manuscript. Colored words below are parts of the text we revised according to your suggestions. Comment 1. Title: considering the small number of animals, also due to the ethical issues related to the 3R, I believe that the title of the work should be changed. It might be worth adding that this is a pilot study. Response 1. Thank you for your comment. According to your suggestion, we corrected the title as follow. Previous Manuscript: title “Safety and osteointegration of titanium screws coated with a fibroblast growth factor-2–calcium phosphate composite layer in non-human primates” Revised Manuscript: title “Safety and osteointegration of titanium screws coated with a fibroblast growth factor-2–calcium phosphate composite layer in non-human primates: A pilot study” Comment 2. Abstract: Even in the abstract it should be emphasized that this is a pilot study. Also, I think it might be useful to introduce the issue of spinal fusion surgery already in the abstract. Response 2. Thank you for your comment. According to your suggestion, we emphasized this is a pilot study and introduced the issue of spinal fusion surgery, and corrected the abstract as follow. Previous Manuscript: Abstract “Abstract: Fibroblast growth factor-2 (FGF-2) promotes bone formation; thus, coating pedicle screws with an FGF-2–calcium phosphate (FGF–CP) composite layer is hypothesized to enhance osteointegration in spinal implants. In this long-term implantation study, we report the safety and bone-forming efficacy of pedicle screws coated with an FGF-CP composite layer in cyno-molgus monkeys. Titanium alloy screws, either uncoated (controls) or aseptically coated with an FGF–CP composite layer, were implanted in the vertebral bodies of six female adult cynomol-gus monkeys (three monkeys per group) for 85 days. Physiological, histological, and radio-graphic investigations were performed. There were no serious adverse events, and no radiolu-cent areas were observed around the screws in either group. The bone apposition rate in the in-traosseous region was significantly higher in the FGF–CP group than in the controls. Moreover, as analyzed by Weibull plots, the bone apposition and bone formation rates of the FGF–CP group exhibited a significantly higher regression line slope than the control group. These results demonstrated that there was significantly less risk of impaired osteointegration in the FGF–CP group. Our data suggest that FGF–CP coated implants could promote osteointegration, be safe, and reduce the probability of screw loosening.” Revised Manuscript: Abstract “Abstract: Spinal instrumentation surgery for older patients with osteoporosis is increasing. Implant loosening may occur due to inappropriate fixation in osteoporotic bone. Developing implants that achieve stable surgical results, even in osteoporotic bone, can reduce re-operation, lower medical costs, and maintain the physical status of older patients. Fibroblast growth factor-2 (FGF-2) promotes bone formation; thus, coating pedicle screws with an FGF-2–calcium phosphate (FGF–CP) composite layer is hypothesized to enhance osteointegration in spinal implants. We designed a long-term implantation pilot study that estimated the safety and bone-forming efficacy of pedicle screws coated with an FGF-CP composite layer in cynomolgus monkeys. Titanium alloy screws, either uncoated (controls) or aseptically coated with an FGF–CP composite layer, were implanted in the vertebral bodies of six female adult cynomolgus monkeys (three monkeys per group) for 85 days. Physiological, histological, and radiographic investigations were performed. There were no serious adverse events, and no radiolucent areas were observed around the screws in either group. The bone apposition rate in the intraosseous region was significantly higher in the FGF–CP group than in the controls. Moreover, as analyzed by Weibull plots, the bone apposition and bone formation rates of the FGF–CP group exhibited a significantly higher regression line slope than the control group. These results demonstrated that there was significantly less risk of impaired osteointegration in the FGF–CP group. Our pilot study’s suggest that FGF–CP coated implants could promote osteointegration, be safe, and reduce the probability of screw loosening.” Comment 3. Introduction: -all the first part of the introduction (lines 32-38) should be implemented, focusing on the clinical problems currently present for this type of surgery. Response 3. Thank you for your comment. According to your suggestion, we added about the clinical problems of the spine instrumentation surgery as follow. Previous Manuscript: 1. Introduction, 1st paragraph “Spinal instrumentation surgery is a useful surgical method for spinal diseases, such as degenerative scoliosis, vertebral fractures, and spinal stenosis. With the growing older adult population, the number of patients with osteoporosis is increasing, along with the use of spinal instrumentation surgery to treat these patients [1,2]. Complications—such as implant loosening, vertebral fracture, and implant failure—frequently occur in patients with osteoporosis [3,4]; therefore, it is important to develop implants that can contribute to stable outcomes, even in older patients with osteoporosis.” Revised Manuscript: 1. Introduction, 1st paragraph “Spinal instrumentation surgery is a useful surgical method for spinal diseases, such as degenerative scoliosis, vertebral fractures, and spinal stenosis. With the growing older adult population, the number of patients with osteoporosis is increasing, along with the use of spinal instrumentation surgery to treat these patients [1,2]. Complications—such as implant loosening, vertebral fracture, and implant failure—frequently occur in patients with osteoporosis [3,4]. Polymethylmethacrylate is one of the materials used to augment the fixation force, but there is a risk of cement extravasation [5]. Therefore, it is important to develop implants that can contribute to stable outcomes and safety, even in older pa-tients with osteoporosis.” Comment 4. -lines 55-57 “clinical applications require preclinical in vivo pathological evaluations of safety and efficacy in non-human primates”. What ISO standard is it? safety, efficacy and osseointegration are always evaluated in small animals. Response 4. Thank you for the comment. ISO standards (ISO 10993 series) on medical devices do not fully cover drug-combination medical devices such as the pedicle screws coated with the FGF-CP layers. Safety and efficacy of drugs are species and tissue dependent. In particular, efficacy and safety of growth factors such as FGF-2 and BMP differs in different species more greatly than low-molecular chemical drugs. Thus, it is preferable to evaluate the efficacy and safety of pedicle screws coated with the FGF-CP layers in addition to the evaluation on the basis of ISO 10993. We have revised the introduction in order to explain the situation mentioned above. Previous Manuscript: 1. Introduction, 3rd paragraph “FGF-2 promotes the cellular differentiation of progenitor cells to osteoblasts and os-teocytes [10]; additionally, it was reported that local application of recombinant human FGF-2 (rh FGF-2) in gelatin hydrogel accelerates bone union [11]. It was also previously reported that ceramic hydroxyapatite coated with FGF–CP composite layers accelerated osteogenesis in a rat model of cranial bone defect and low-dose rhFGF-2 also increased osteogenesis [12]. Further, titanium screws coated with FGF–CP composite layers have been shown to reduce the risk of impaired bone apposition to the screw [13]. However, these previous animal studies used non-primate species. Since the optimal dose of rhFGF-2 for osteogenesis may differ between these species and primates, clinical applica-tions require preclinical in vivo pathological evaluations of safety and efficacy in non-human primates.” Revised Manuscript: 1. Introduction, 3rd paragraph “FGF-2 promotes the cellular differentiation of progenitor cells to osteoblasts and osteocytes [23]; additionally, it was reported that local application of 0.8 and 2.4 mg of recombinant human FGF-2 (rh FGF-2) in gelatin hydrogel accelerates bone union in human [24]. It was also previously reported that ceramic hydroxyapatite coated with FGF–CP composite layers accelerated osteogenesis in a rat model of cranial bone defect and low-dose rhFGF-2 also increased osteogenesis [25]. Further, titanium screws coated with FGF–CP composite layers have been shown to reduce the risk of impaired bone apposition to the screw in a rabbit model, in which the titanium screws were loaded with 2.0-4.7 µg of rh FGF-2 in Bradford assay [26]. However, these previous animal studies used non-primate species. In non-primate species, rh FGF-2 is a foreign body that can induce safety and efficacy pro-files that are different from those in primate species. In addition, an optimal dose of growth factor depends on the species, tissue, and drug carriers (such as gelatin and CP). Thus, it is preferable to carry out in vivo pathological evaluations of safety and efficacy of pedicle screws coated with FGF–CP composite layers in non-human primates in the same tissues.” Comment 5. Materials and Methods Animal experiments: Lines 122: ‘Three screws were implanted into three vertebral bodies of each animal. The surgical wound was closed in a standard manner’. Please explain. Response 5. Thank you for your comment. According to your suggestion, we corrected the sentence as follow. Previous Manuscript: 2.4 Animal experiments, 3rd paragraph “A left anterior retroperitoneal approach was used, and the lateral side of the lumber vertebral bodies was exposed. Three screws were implanted into three vertebral bodies of each animal. The surgical wound was closed in a standard manner, and the monkeys were allowed to graze immediately after surgery without external immobilization. Health and behavioral observations, blood tests, and radiography were performed 1 week before the operation and on days 8 (1 week after the operation), 29, 57, and 85. Adverse events assumed to be complications of surgery included findings of infection, inflammation due to an allergic reaction to the FGF–CP composite layer, remarkable deterioration of health, and the occurrence of new diseases and injuries.” Revised Manuscript: 2.4 Animal experiments, 3rd paragraph “A left anterior retroperitoneal approach was used, and the lateral side of the lumber vertebral bodies was exposed. Three screws were implanted into three vertebral bodies of each animal. The surgical wound was closed fascia sutured, and the subcutaneous tissue sutured with absorbable sutures, and the monkeys were allowed to graze immediately after surgery without external immobilization. Health and behavioral observations, blood tests, and radiography were performed 1 week before the operation and on days 8 (1 week after the operation), 29, 57, and 85. Adverse events assumed to be complications of surgery included findings of infection, inflammation due to an allergic reaction to the FGF–CP composite layer, remarkable deterioration of health, and the occurrence of new diseases and injuries.” Comment 6. Preparation of tissue specimens: Line 148: ‘non-decalcified hard tissue specimens were prepared.’ How? Please explain the hard resin used and how you obtain 40 µm sections. Response 6. Thank you for your comment. According to your suggestion, we corrected the sentences as follow. Previous Manuscript: 2.7 Preparation of tissue specimens “The animals were euthanized on day 85 using 120 mg/kg pentobarbital (Somno-pentyl; Kyoritsu Seiyaku Corporation, Tokyo, Japan). The vertebral bodies and screws were fixed in 10% neutral-buffered formalin, and non-decalcified hard tissue specimens were prepared. Vertebral body sections (40 μm) were obtained parallel to the screw hole and across the center of the screw, and cortical bone mineral density (BMD), trabecular BMD, and cortical thicknesses were measured by microcomputed tomography. After pol-ishing to exclude scratches, the sections were stained with hematoxylin-eosin (HE) and toluidine blue (TB). The spinal cord and soft tissue around the screw head were then fixed with 10% neutral-buffered formalin and embedded in paraffin. Further, the embedded samples were sliced into 5-μm sections and stained with HE and Masson’s trichrome. The sections were randomized and blinded.” Revised Manuscript: 2.7 Preparation of tissue specimens “The animals were euthanized on day 85 using 120 mg/kg pentobarbital (Somno-pentyl; Kyoritsu Seiyaku Corporation, Tokyo, Japan). The vertebral bodies and screws were fixed in 10% neutral-buffered formalin, embedded in methyl methacrylate resin, and non-decalcified hard tissue specimens were prepared. Vertebral body sections (40 μm) were obtained parallel to the screw hole and across the center of the screw by micro-cutting machine (BS-300CP MEIWAFOSIS Corporation, Tokyo, Japan). After polishing to ex-clude scratches, the sections were stained with hematoxylin-eosin (HE) and toluidine blue (TB). The spinal cord and soft tissue around the screw head were then fixed with 10% neutral-buffered formalin and embedded in paraffin. Further, the embedded samples were sliced into 5-μm sections and stained with HE and Masson’s trichrome. The sections were randomized and blinded.” Comment 7. Line 151: microcomputed tomography. Please explain the acquisition and reconstruction methods. I think a dedicated paragraph would be needed. Response 7. Thank you for your comment. We used microcomputed tomography to measured bone mineral density of vertebral body. We haven't used it except to measure bone mineral density. Therefore, we added a sentence as follow. Previous Manuscript: 2.6 Radiography “Anteroposterior and lateral radiography of the screws was performed on days 8, 29, 57, and 85. Moreover, three blinded orthopedic physicians evaluated the radiographs. On anteroposterior radiographs, a wide (>1 mm) radiolucent zone surrounding the screws, regardless of the length of lucency, was defined as radiographic loosening.” Revised Manuscript: 2.6 Radiography “Anteroposterior and lateral radiography of the screws was performed on days 8, 29, 57, and 85. Moreover, three blinded orthopedic physicians evaluated the radiographs. On anteroposterior radiographs, a wide (>1 mm) radiolucent zone surrounding the screws, regardless of the length of lucency, was defined as radiographic loosening. Bone mineral density was measured at an unscrewed vertebral body by micro-computed tomography (LCT-100A, Nihon Ray Tech Corporation, Tokyo, Japan).” Comment 8. Histological evaluation of the bone around the screw Have you calculated the BIC? Figure 1: is it a scan or histology? I believe it is a macro scan Figure 3: Histological sections are not so clear. What about micro-CT results? Response 8. Thank you for your comment. We have not calculated the BIC. Figure 1 is not a micrograph, but an overall image taken with a digital camera. It is shown to explain the positional relationship between the inserted screw and the vertebral body and the ROI. According to your suggestion, we replaced Figure 3. Micro CT results are bone mineral density (g/cm2). Those are described in Tables 2 and the first paragraph of 3.2 Animal study. Comment 9. Discussion Line 303: ‘medical product’. Please replace with medical device. Line 310: Please add references. I believe that given the low number of animals used, the discussion should be modified, trying to underline that the results are preliminary and that their value is in relation to the number of animals. The use of 3 animals per experimental group does not allow us to be so conclusive. Thus, the discussion should be substantially modified. Response 9. Thank you for your comment. According to your suggestion, we changed from “medical product“ to “medical device”, and added reference (35). According to your suggestion, considering that the number of animal experiments used was small, we have revised the assertive expression in the Discussion as follow. Previous Manuscript: 4. Discussion, 2nd paragraph “The present study demonstrated that 4.0 μg/ml of FGF-2 in a supersaturated calcium phosphate solution results in the formation of FGF–CP composite layers that potentially promote osteointegration in primates. Previous studies have examined the effectiveness of FGF–CP composite layers in rabbits and rats, where the same concentration of FGF-2 (4.0 μg/ml) was shown to exert the most beneficial osteogenic effect [12-13, 26-28]. In this study, titanium alloy screws coated with FGF–CP composite layers prepared using the same concentration of FGF-2 also promoted bone apposition to the screws and reduced the risk of impaired bone apposition and formation when compared with uncoated screws in primates. The promoting effect of FGF–CP composite layers on osteointegration results from both calcium phosphate and FGF-2. Nonetheless, the dose of FGF-2 in the FGF–CP composite layer in this study is likely to contribute to osteointegration and remain within the effective dose range in primates. It should be noted that the probability of impaired bone apposition to uncoated titanium screws was similar to that of calcium phos-phate-coated screws without FGF-2 in a previous study [13]. These results suggest safety and osteointegration based on pathological findings and clinical trials using spinal screws coated with FGF–CP composite layers. Moreover, we observed promising results when the layers were coated under the same conditions.” Revised Manuscript: 4. Discussion, 2nd paragraph “The present study demonstrated that 4.0 μg/ml of FGF-2 in a supersaturated calcium phosphate solution results in the formation of FGF–CP composite layers that potentially promote osteointegration in primates. Previous FGF–CP composite layers prepared under similar conditions contained poorly crystalline apatite or amorphous calcium phosphate and had a Ca/P molar ratio in the range of 1.55-1.85 [40-43]. Since the Ca/P molar ratio (1.71 ± 0.02, Table 1) in this study fell within similar range to the previous study, poorly crystalline apatite or amorphous calcium phosphate could also be formed in the present study. Previous studies have examined the efficacy of FGF–CP composite layers in rabbits and rats, where the same concentration of FGF-2 (4.0 μg/ml) was shown to exert the most beneficial osteogenic effect [25,26, 40,41,44]. In this pilot study, it was suggested that titanium alloy screws coated with FGF–CP composite layers prepared using the same concentration of FGF-2 also promoted bone apposition to the screws and reduced the risk of impaired bone apposition and formation when compared with uncoated screws in primates. The promoting effect of FGF–CP composite layers on osteointegration results from both calcium phosphate and FGF-2. Nonetheless, the dose of FGF-2 in the FGF–CP composite layer in this study is likely to contribute to osteointegration and remain within the effective dose range in primates. It should be noted that the probability of impaired bone apposition to uncoated titanium screws was similar to that of calcium phosphate-coated screws without FGF-2 in a previous study [26]. These results suggest safety and osteointegration based on pathological findings and clinical trials using spinal screws coated with FGF–CP composite layers. Moreover, we observed promising results when the layers were coated under the same conditions.” Previous Manuscript: 4. Discussion, 4th paragraph “This study of FGF–CP-coated spinal screw implants in cynomolgus monkeys demon-strated local safety based on a long-term in vivo exam. Neither malignant cells nor infec-tion-mediated inflammatory cell infiltration was observed around the coated screws. A previous study reported that gamma-ray-sterilized screws coated with FGF-CP were safe in cynomolgus monkeys [18]. In this study, no infiltration of inflammatory cells scored 3 or 4 in ISO 109993-6 were observed in the spinal cord and vertebral bone around the coat-ed and uncoated screws. Further, no adverse events—excess osteogenesis in the spinal canal or motor paralysis—were observed; however, some implants penetrated the spinal canal. The FGF-2 on the implant surface may have been released into the spinal canal. Moreover, although bone and fibrous tissue could be induced into the spinal canal by the effect of FGF-2 on the implant surface, histopathology performed 85 days after implanta-tion suggested that osteointegration could be induced without causing abnormal mor-phology of the spinal cord and nerve tissue.” Revised Manuscript: 4. Discussion, 5th paragraph “This pilot study of FGF–CP-coated spinal screw implants in cynomolgus monkeys demonstrated tentative local safety based on a long-term in vivo exam. Neither malignant cells nor infection-mediated inflammatory cell infiltration was observed around the coated screws. A previous study reported that gamma-ray-sterilized screws coated with FGF-CP were safe in cynomolgus monkeys [31]. In this study, no infiltration of inflammatory cells scored 3 or 4 in ISO 109993-6 were observed in the spinal cord and vertebral bone around the coated and uncoated screws. Further, no adverse events—excess osteogenesis in the spinal canal or motor paralysis—were observed; however, some implants penetrated the spinal canal. The FGF-2 on the implant surface may have been released into the spinal canal. Moreover, although bone and fibrous tissue could be induced into the spinal canal by the effect of FGF-2 on the implant surface, histopathology performed 85 days after implantation suggested that osteointegration could be induced without causing abnormal morphology of the spinal cord and nerve tissue.” Previous Manuscript: 5. Conclusions “Long-term implantation tests on FGF–CP coated spinal screws confirmed their safety in non-human primates. The screws induced significantly higher osteointegration and a remarkably lower probability of impaired osteointegration than uncoated screws. Our da-ta suggested that FGF–CP coated spinal screws could reduce the probability of screw loosening and contribute to stable outcomes in spinal instrumentation surgery. However, further studies are required to verify the results of our study.” Revised Manuscript: 5. Conclusions “This pilot study of long-term implantation tests on FGF–CP coated spinal screws confirmed their safety in non-human primates. Tentatively, the screws induced significantly higher osteointegration and a remarkably lower probability of impaired osteointegration than uncoated screws. Our data suggests that FGF–CP coated spinal screws could reduce the probability of screw loosening and contribute to stable outcomes in spinal instrumentation surgery. However, further studies are required to verify the results of our study.”
Round 2
Reviewer 2 Report
The manuscript has been greatly improved and the authors have answered all questions. I think that the manuscript can be published in the present form.